# NEURAL CAUSAL REGULARIZATION UNDER THE INDEPENDENCE OF MECHANISMS ASSUMPTION

**Mohammad Taha Bahadori**[1]**, Krzysztof Chalupka**[2]**, Edward Choi**[1]**,
Robert Chen**[1]**, Walter F. Stewart**[3]**, & Jimeng Sun**[1]

[1] Georgia Institute of Technology,     [2] California Institute of Technology,     [3] Sutter Health

## ABSTRACT

Neural networks provide a powerful framework for learning the association between input and response variables and making accurate predictions and offer promise in using the rapidly growing volume of health care data to surface causal relationships that cannot necessarily be tested in randomized clinical trials. In pursuit of models whose predictive power comes maximally from causal variables, we propose a novel causal regularizer based on the independence of mechanisms assumption. We use the causal regularizer to steer deep neural network architectures towards causally-interpretable solutions. We perform a large-scale analysis of electronic health records. Our causally-regularized algorithm outperforms its $L_1$-regularized counterpart both in predictive performance as well as causal relevance. Finally, we show that the proposed causal regularizer can be used together with representation learning algorithms to yield up to $20\%$ improvement in the causality score of the generated hypotheses.

## 1 INTRODUCTION

In domains such as healthcare, genomics or social science there is high demand for data analysis that reveals *causal* relationships between independent and target variables. For example, doctors not only want models that accurately predict the status of patients, but also want to identify the factors that can change the status. The distinction between prediction and causation has at times been subject to controversy in statistics and machine learning (Breiman et al., 2001; Shmueli, 2010; Donoho, 2015). On one hand, machine learning has been focusing almost exclusively on pure prediction tasks, enjoying great commercial success. On the other hand, in many scientific domains pure prediction without consideration of the underlying causal mechanisms is considered unscientific (Shmueli, 2010). In this work, we propose a neural causal regularizer that balances causal interpretability and high predictive power.

**Causal Inference:** Our notion of causality follows the counterfactual framework of Pearl (2000). Thus, we will say that one random variable $X$ *causes* another variable $Y$ (which relationship we denote as $X \to Y$) if *intervening* or *experimenting* on $X$ changes the distribution of $Y$. Consider the problem of identifying the causal relationship between drinking red wine and heart disease (Spirtes, 2010). Regular consumption of red wine correlates with healthy heart. That might mean that drinking red wine decreases heart attack rates. But it might be, for example, that people of high socio-economic status tend to drink more wine, while at the same time tend to suffer fewer heart problems due to better living conditions. To distinguish between these two possibilities, one could implement a controlled trial in which the subjects are told to drink (or not drink) red wine, independently of *any other factors* —including their socio-economic status.

Such controlled trials are often undesirable or even impossible. In healthcare, it can be due to moral and regulatory reasons; in climate science for example, due to technological limitations (we don't know how to change climate). In such settings, we would like to still establish causality without resorting to experiment. Even in applications where controlled trials are possible, the large number of causal hypotheses can make it impossible to experimentally test all of them. Furthermore, in domains such as healthcare, many causal factors need to occur simultaneously to have an effect on the target variable, a scenario that we call *multivariate causation*. Given the exponential number of combinations of the independent variables and different transformations, it is even more difficult to explore all of these multivariate causation scenarios.

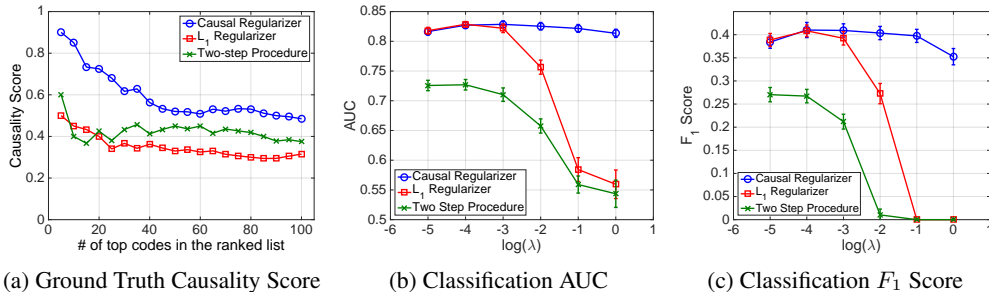

Figure 1: Superior causality and predictive performance of the causal regularizer in our heart failure study. (a) Average causality score computed using ground truth causality labels. We compute the score for top $k$ codes reported by three algorithms. (b and c) The causal regularizer is more stable in predictive performance measured by AUC and $F_1$ metrics. See Section 3 for more details.

Analyzing causation without resorting to experiment is challenging due to unobserved confounders (such as the possible influence of socio-economic status on heart health and wine drinking). Many methods have been proposed for discovering causal relationships among multiple variables from observational data only (Chickering, 2002; Kalisch & Bühlmann, 2007; Colombo et al., 2012), demonstrating various degrees of success. These methods are based on the idea that any given set of causal relationships among multiple variables will leave in the joint distribution well-defined markers in the form of independence relationships among subsets of the variables. These methods, however, are often very sensitive to small changes in the joint distribution.

**Causal Regularization:** Our main idea is to design a *causal regularizer* to control the complexity of the statistical models and at the same time favor causal explanations. Compared to the two step procedure of (i) causal variable selection and (ii) multivariate regression, the proposed approach performs joint causal variable selection and prediction, thus avoiding the statistically sensitive hard-thresholding of the causality scores in the causal variable selection step. It allows dependencies that cannot be explained via causation to be included in the model. Our contributions are four-fold:

1. We propose a customized causality detector neural network that can accurately discriminate causal and non-causal variables in our healthcare datasets. To this end, we propose new synthetic dataset generation to train the causal structure detectors in (Chalupka et al., 2016; Lopez-Paz et al., 2016) with additional prior knowledge from the healthcare domain.
2. We use the causality detector to construct a causal regularizer that can guide predictive models towards learning causal relationships between the independent and target variables.
3. Given the fact that the causal regularizer seamlessly integrates with non-linear predictive models such as neural networks, we propose a new non-linear predictive model regularized by our causal regularizer, which allows *neural causally predictive modeling*.
4. Finally, we demonstrate that the proposed causal regularizer can be combined with neural representation learning techniques to efficiently generate multivariate causal hypotheses.

The proposed framework scales linearly with the number of variables, as opposed to many previous causal methods. Combined with a predictive model, it efficiently screens a high-dimensional hypothesis space and proposes plausible hypotheses.

We applied the proposed algorithm to two electronic health records (EHR) datasets: Sutter Health's heart failure study data and the publicly available MIMIC III (Johnson et al., 2016) dataset. Altogether, we analyzed the influence of 17,081 independent variables on heart failure. To validate our claims, we use expert judgment as the causal ground-truth to compare our causal-predictive solutions with purely predictive solutions that do not take causality into account. As shown in Figure 1, a causally-regularized algorithm outperforms its $L_1$-regularized equivalent both in predictive performance as well as causal performance.

## 2 METHODOLOGY

In order to "inject causality" into predictive models, we use the "independence of cause and mechanisms" (ICM) assumption, which allows us to construct a *neural network causality detector*, as described in Section 2.1. We present a causal regularizer for linear models in Section 2.2. Using this regularizer, in Section 2.3 we propose non-linear deep neural networks to learn non-linear causal relationships between the independent and target variables. Finally, we show that the causal regularizer can efficiently explore the space of multivariate causal hypotheses and extract meaningful candidates for causality analysis.

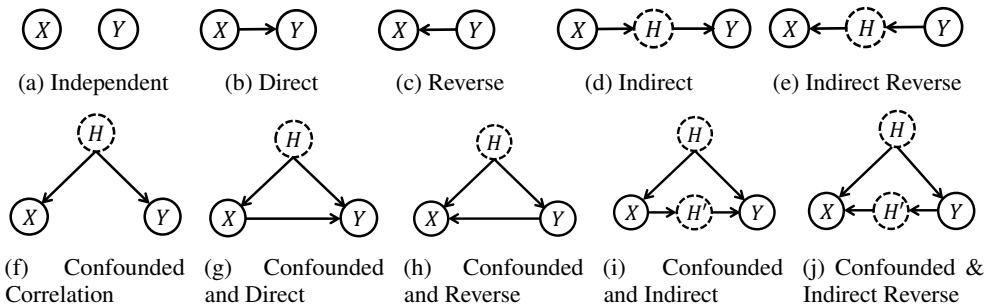

(a) Independent (b) Direct (c) Reverse (d) Indirect (e) Indirect Reverse

(f) Confounded Correlation (g) Confounded and Direct (h) Confounded and Reverse (i) Confounded and Indirect (j) Confounded & Indirect Reverse

Figure 2: Some possible causal structures between two observed and one or more hidden variables. Under the algorithmic independence assumption, we can sample from the joint distribution of $X$ and $Y$ in each case and train a classifier that distinguishes between these cases based on the (automatically learned) features of the joint distribution.

## 2.1 CAUSALITY DETECTION BASED ON INDEPENDENCE OF MECHANISMS

As we discussed in the introduction, the task of analysis of causal effect of multiple independent variables on a target variable is difficult. Our approach in this paper is to reduce the problem to analysis of the causal effect of a single independent variable $X$ on the target variable $Y$, which is known as pairwise causal analysis. In the next subsections, we will describe how to use a pairwise causality detector to perform multivariate causality analysis.

In particular, we are interested in finding causal models where $X$ causes $Y$, or $Y$ causes $X$, or the two are confounded based on joint distribution of $P(X, Y)$. However, even the pairwise causality analysis is infeasible for arbitrary joint distributions. Thus, we need to resort to additional assumptions on the nature of the causal relationships. Recently several algorithms have been proposed that distinguish between the cause and effect based on the natural assumption that steps in the process that generates the data are independent from each other, see (Lemeire & Dirkx, 2006; Janzing et al., 2012; Daniusis et al., 2010; Lopez-Paz, 2016; Chalupka et al., 2016) and the references therein. In this work, we follow (Lopez-Paz et al., 2016; Chalupka et al., 2016) to describe this causality detection approach. In the next subsections, we describe our novel causal regularizer designed based on this causality detection approach and its application in non-linear causality analysis and multivariate causal hypothesis generation.

**Conceptual description of the independence between the cause and the mechanism.** Algorithms based on the ICM, such as (Chalupka et al., 2016; Lopez-Paz et al., 2016) do not put assumptions on the functional form of the causal relationships between the variables of interest. Instead, they are based on the following assumption on how causal mechanisms come to be. ICM states that the two processes of generation of the cause and mapping from cause to effect are in some sense independent. In our case, we assume that when $X \rightarrow Y$ ($X$ causes $Y$), the probabilities $\mathbb{P}(Y \mid X)$ and $\mathbb{P}(X)$ are generated by independent higher-level distributions. This conforms to the scientific idea of Uniformitarianism (Gould, 1965) which, putting roughly, states that the laws of nature apply to all objects similarly. ICM can be described in both deterministic (Janzing & Scholkopf, 2010) and probabilistic sense (Daniusis et al., 2010); this work mainly uses the probabilistic interpretation.

ICM can be used to generate all of the possible graphical models including two observed variables $X$ and $Y$ and an unobserved variable $H$ shown in Figure 2, by requiring that the probabilities in the factorization are independent from each other. The hidden variables can represent the other observed variables such as $Z$, critical in design of the regularizer in the next subsection.

Following the ICM, we assume that each cause-effect link in the world is probabilistic and can be described by a joint distribution $P(cause, effect)$. In addition, the link itself is sampled from a probabilistic *hyperprior*. The key assumption is on the structure of this hyperprior, namely that it decomposes into two parts $\Pi_c$ and $\Pi_m$ that have the following properties:

1. For each $cause, effect$ pair, Nature samples the cause's distribution $P_{cause}$ from a *hyperprior* $\Pi_c[P_{cause}]$.

---

Given the data, perform the following steps:

1. Generate data samples $S_i$ for $i = 1, \ldots, n_{\text{train}}$ from $p_{X,Y}$ according to the ten cases in Figure 2.

2. Assign label $y = 0$ to the cases in Figures 2b, 2d, 2g and 2i and $y = 1$ to the rest.

3. Train a classifier $f : \mathcal{S} \to [0, 1]$ to classify them as causation (label=1) or not-causation (label=0). Given the fact that this is a synthetic dataset, we know these labels and we can use supervised learning.

4. On the test set, construct the test sample sets and use the classifier in step 3 to classify the example.

---

Algorithm 1: The algorithm for constructing the causality detector. The structure of neural network classifier is given in Appendix B.1.

2. At the same time, Nature samples the causal mechanism (the distribution of the effect conditioned on the cause) $P_{effect|cause}$ from a *hyperprior* $\Pi_m[P_{effect|cause}]$.
3. The hyperpriors are flat —for discrete *cause* and *effect*, they are Dirichlet distributions with $\alpha$ uniformly equal to 1.

The last assumption is not crucial and can easily be changed if knowledge about hyperpriors in a specific domain is available. In fact, we tailor the hyperpriors to our task below. These three assumptions give us a full generative model of causal links in the world, a model under which the likelihood ratio test can be used to differentiate between the data generated from each of the ten cases shown in Fig. 2. Chalupka et al. (2016) developed an analytical likelihood ratio test that decides between the causal and anticausal cases (Figures 2b and 2c). Taking into account the confounded cases is, however, difficult or impossible to compute analytically. Nevertheless, it is possible to generate samples from the generative model defined by the ICM and train a neural network to learn to choose the max likelihood causal structure given samples from the joint $P(cause, effect)$. This is the key idea of the causality detectors in (Lopez-Paz et al., 2016; Chalupka et al., 2016).

**Mathematical description of the causality detection algorithm.** Formally, suppose we have $m$ variables $X_i$, each with dimensionality $d_i$. For each variable we observe a sample of size $n_i$ denoted by $S_i = \{(\mathbf{x}_{i,j}, y_j)\}_{j=1}^{n_i}$, where $y_j$ are observations of a common target variable $Y$. Let $\mathcal{S}$ denote the set of all such samples. For each sample $S_i$, we are interested in determining the binary label $\ell_i \in \{0, 1\}$ which determines whether $X_i$ causes $Y$ or not. In fact, we are interested in the function approximation problem of learning the mapping $f : \mathcal{S} \mapsto \{0, 1\}$.

Several approaches can learn such a mapping function. When $X$ and $Y$ are both discrete and finite, Chalupka et al. (2016) construct the empirical joint distribution $\widehat{p}_i = \widehat{p}(X_i, Y)$ and train a supervised neural network mapping function $f(\widehat{p}_i) \to \ell_i$. Lopez-Paz et al. (2016) learn the mapping $\frac{1}{n_i} \sum_{j=1}^{n_i} \phi(\mathbf{x}_{i,j}, y_j)$ and a neural network $f\left(\frac{1}{n_i} \sum_{j=1}^{n_i} \phi(\mathbf{x}_{i,j}, y_j)\right) \to \ell_i$. They train both the representation leaning function $\phi(\cdot, \cdot)$ and the classification network in a joint and supervised way.

However, it is rare to have the true causal labels $\ell$ for training a causal detector. The key idea is to generate a synthetic dataset composed of the cases in Figure 2 based on the ICM assumption. As shown in Algorithm 1, the overall procedure is to generate samples from distributions $p_{X,Y}$ that are one of the ten possible cases in Figure 2. We need to select the distributions such that they impose minimum number of restriction on the data and the synthetically-generated distributions have statistics as similar as possible to those of our true data of interest. For example, in our dataset, the independent variables $X$ are counts of the number of disease codes in patients' records (cf. Section 3). Thus, we sample $X$ from a mixture of appropriate distributions for count data: the Zipf, Poisson, Uniform, and Bernoulli distributions. The hidden variable $H$ and the response variable $Y$ are sampled from the Dirichlet and Bernoulli distributions. Details of our sampling procedure are provided in Appendix A.

## 2.2 THE CAUSAL REGULARIZER

As an instructive alternative to our approach, consider the two-step analysis method of first finding the variables $X_i$ that are most likely causes of $Y$ and then performing a sparse multivariate regression to select the important variables. Ideally, if the ICM holds and if we had access to the true joint distributions and could discriminate between causal and non-causal variables with perfect accuracy, the two-step procedure would be sufficient. But real-world datasets always contain noise and selection bias, which can perturb the causality scores generated by the neural network confounder detector.

The problem arises from the fact that our causality detection algorithm might give soft scores such as $0.5 + \varepsilon$ or $0.5 - \varepsilon$ to two variables $X_1$ and $X_2$, respectively. These soft-scores can be interpreted as the probability that each variable is the cause of $Y$. If we use the two-step procedure, we will include $X_1$ in the regression model but not $X_2$. However, $X_2$ could possibly contribute more to the predictive performance in presence of other variables in the multivariate regression. In other words, any hard cut-off for the purpose of two-step causal variable selection and regression will pose the question of "what should be the best cut-off threshold?" Note that any hard cut-off will be always statistically unstable in presence of noise and selection bias.

Instead, we propose a causally regularized regression approach, where this trade-off is performed naturally via a regularization parameter. We select variables that are both potentially causal with high probability and also significantly predictive.

**Causal Regularizer.** Now that we have a classifier that outputs $c_i = \mathbb{P}[X_i$ and $Y$ are not-causal], we can design the following regularizer to encourage learning a causal predictive model:

$$\widehat{\mathbf{w}} = \underset{\mathbf{w}}{\operatorname{argmin}} \left\{ \frac{1}{n} \sum_{j=1}^{n} \mathcal{L}(\mathbf{x}_j, y_j | \mathbf{w}) + \lambda \sum_{i=1}^{m} c_i |w_i| \right\}, \qquad (1)$$

where $\mathcal{L}(X_1, \ldots, X_n, Y | \mathbf{w})$ is the loss function of logistic regression for $X_1, \ldots, X_n$ and $Y$. The first term in Eq. (1) is a multivariate analysis term, whereas the regularizer might look like a bivariate operation between each independent variable $X_i$ and the target variable $Y$ for $i = 1, \ldots, p$. However, we should note that in the design of the causal regularizer, we have implicitly included the other variables as hidden variables in the analysis. Thus we are allowed to use the regularizer together with multivariate regression. Note that the proposed causal regularizer is also a decomposable regularizer which makes analysis of its theoretical properties easier (Negahban et al., 2012).

The two-step analysis can be cast as a special case of causally predictive modeling where we use hard scores instead of soft scores. Consider the following setting:

$$\widehat{\mathbf{w}} = \underset{\mathbf{w}}{\operatorname{argmin}} \left\{ \frac{1}{n} \sum_{j=1}^{n} \mathcal{L}(\mathbf{x}_j, y_j | \mathbf{w}) + \gamma \sum_{i=1}^{m} c_i' |w_i| \right\},$$

Where $c_i'$ is defined as follows:

$$c_i' = \begin{cases} 1 - \varepsilon & \text{if } c_i > 1/2 \\ \varepsilon & \text{if } c_i \leq 1/2 \end{cases}$$

Now, consider the limiting case of $\varepsilon \to 0$ and $\gamma \varepsilon \to \lambda$. This case corresponds to the two-step procedure with $L_1$ regularized logistic regression.

Note that the possibility of having a causal regularizer has been proposed in (Lopez-Paz, 2016, Page 181) and (Lopez-Paz et al., 2016), however a specific causal regularizer has never been developed and evaluated. Furthermore, note that using the score of a "causal-anticausal"-only classifier, as e.g. in (Lopez-Paz et al., 2016), cannot properly regularize a multivariate model such as logistic regression. In our proposal, the rest of the observed independent variables can be considered as hidden variables in our bivariate causality analysis which allows proper regularization. Moreover, a major novelty of our proposed causal regularizer is to do joint causal variable selection (the $L_1$ regularization) and prediction, but the idea in (Lopez-Paz et al., 2016) cannot.

## 2.3 CAUSAL REGULARIZERS IN NEURAL NETWORKS

The key advantages of causal regularizer can be seen when it is used for regularizing neural networks. We demonstrate two use cases of causal regularizer as shown in Figure 3.

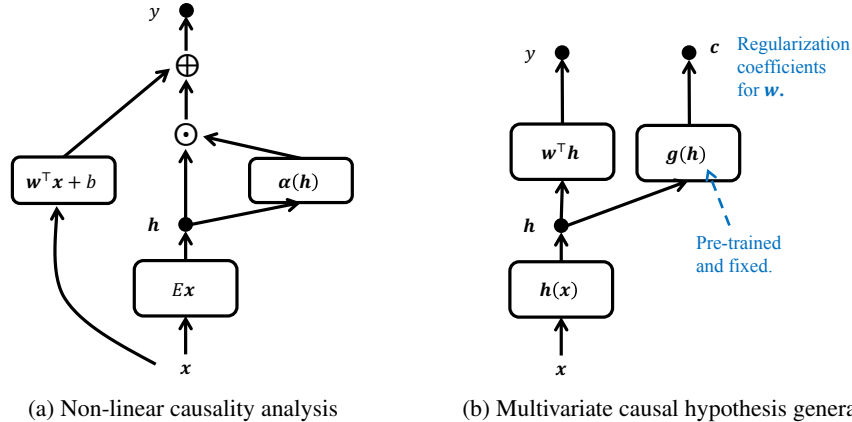

(a) Non-linear causality analysis (b) Multivariate causal hypothesis generation

Figure 3: Two use cases of the proposed causal regularizer: (a) In the proposed architecture, applying the causal regularizer allows identification of causal relationships in the non-linear settings, where the causality coefficient can change from subject to subject. (b) The causal regularizer allows us to explore the high-dimensional multi-variate combinations of the variables and identify plausible hypotheses. Here, $g$ generates the causal regularization coefficients for the hypotheses $h$. The regularizer encourages the coordinates of $h$ to be more causal.

**Non-linear Modeling.** The objective is to design a non-linear neural network in a way that we can still identify causality. We propose the following non-linear generalized linear model:

$$\sigma^{-1}(\mathbb{E}[Y]) = \mathbf{w}^\top \mathbf{x} + \boldsymbol{\beta}^\top (\boldsymbol{\alpha}(E\mathbf{x}) \odot (E\mathbf{x})) + b, \tag{2}$$

where the embedding matrix $E \in \mathbb{R}^{q \times m}$ maps the input $\mathbf{x} \in \mathbb{R}^m$ to a lower dimensional representation space and the symbol $\odot$ denotes the element-wise product. The logistic sigmoid function $\sigma^{-1}$ maps the real values to the $[0,1]$ interval. The term $\mathbf{w}^\top \mathbf{x}$ acts as the skip connection and initialized by the result of logistic regression. The embedding allows dealing with very large set of discrete concepts and can be initialized via techniques such as skip-gram (Mikolov et al., 2013) or GloVe (Pennington et al., 2014). The vector $\boldsymbol{\alpha}(E\mathbf{x})$ can be computed using a multi-layer preceptron.

The model in Eq. (2) is a particular non-linear extension of logistic regression. We can reorder the equations to write the right hand side of Eq. (2) as $\boldsymbol{\omega}(\mathbf{x})^\top \mathbf{x} + b$, where the new regression coefficient $\boldsymbol{\omega}$ can change with every input. Each coordinate of the new regression coefficient can be calculated as $\omega_i(\mathbf{x}) = w_i + (\boldsymbol{\beta} \odot \boldsymbol{\alpha}(E\mathbf{x}))^\top E_i$, where $E_i$ denotes the $i$th column of the embedding matrix $E$. The variability of $\omega_i(\mathbf{x})$ for each input $\mathbf{x}$ enables us to perform individual causality analysis. For training, we can penalize the $\boldsymbol{\omega}$ coefficients and minimize the following loss function

$$\frac{1}{n} \sum_{j=1}^{n} \left\{ \widetilde{\mathcal{L}}(\mathbf{x}_j, y_j) + \lambda \sum_{i=1}^{m} c_i |\omega_i(\mathbf{x}_j)| \right\}, \tag{3}$$

where $\widetilde{\mathcal{L}}$ denotes the negative log-likelihood of the model described by Eq. (2). The change of the prediction vector with each sample $\mathbf{x}$ can be related to the probabilistic definition of causation (Pearl, 2000) in the sense that the strength of causality may change from a subject to another one. The fact that in Eq. (2) the impact of each independent variable on the target is measured by $\omega_i(\mathbf{x})$ allows us to penalize it with our regularizer and push the model to learn more causal relationships.

**Multivariate Causal Hypothesis Generation.** A key application of our proposed causal regularizer in conjunction with deep representation learning is to efficiently extract multivariate causal hypotheses from the data. Figure 3b shows an example of causal hypothesis generation where the hypotheses are generated via a Multilayer Perceptron (MLP). We assume that there is a representation learning network with $K$-dimensional output $\boldsymbol{h}(\mathbf{x}) \in \mathcal{I}^K$, where $\mathcal{I}$ denotes the range of the output, for example $\mathcal{I} = (0, 1)$ for sigmoid and $\mathcal{I} = [0, \infty)$ for ReLU activation functions. Our goal is to force each dimension of $\boldsymbol{h}$ to be causal, thus $\boldsymbol{h}$ can be used as multivariate causal hypotheses. In particular, we aim at minimizing the following objective function:

$$\frac{1}{n} \sum_{j=1}^{n} \left\{ \mathcal{L}(\mathbf{w}^\top \boldsymbol{h}_j + b) + \lambda \sum_{i=1}^{K} |g_i(\boldsymbol{h}_{j,i}) w_i| \right\} \tag{4}$$

Our approach is to train a causality detector based on (Lopez-Paz et al., 2016) and design the regularizer $g(h(\mathbf{x}))$ based on its score. Then, as shown in Figure 3b, we can combine it with the neural network to regularize the coefficients of the last layer of the multilayer Perceptron which predicts the labels from $h$. The weights of the lower layers in $h(\mathbf{x})$ are regularized using $L_1$ regularizer to make the generated causal variables simple. To train the network, we select batches with fixed-size of 200 examples. This number is selected to be large enough such that error rate of the causality detector in (Lopez-Paz et al., 2016) becomes lower than 2%. We select the non-linearity for $h$ to be the logistic sigmoid function, thus we use Beta distribution for generating synthetic data for training of the causality classifier.

## 3 EXPERIMENTS

We evaluate the proposed causal regularizer in Section 2.2 both in terms of their predictive and causal performance. Next, we compare the quality of the codes identified as causes of heart failure identified by different approaches. Finally, we evaluate performance of multivariate causal hypothesis generation by qualitatively analyzing the extracted hypotheses. We defer evaluation of the causality detection algorithms to Appendix A, as they are not the main contributions of this work.

### 3.1 DATA

The **Sutter Health heart failure dataset** consists of Electronic Health Records of middle-aged adults collected by Sutter Health for a heart failure study. From the encounter records, medication orders, procedure orders and problem lists, we extracted visit records consisting of diagnosis, medication and procedure codes. We denote the set of such codes by $\mathcal{C}$. Given a visit sequence $\mathbf{v}_1, \ldots, \mathbf{v}_T$, we try to predict if the patient will be diagnosed with heart failure (HF) and identify the key causes of increase heart failure risk. To this end, 3,884 cases are selected and approximately 10 controls are selected for each case (28,903 controls). The case/control selection criteria are fully described in the supplementary section. Cases have index dates to denote the date they are diagnosed with HF. Controls have the same index dates as their corresponding cases. We extract diagnosis codes, medication codes and procedure codes from the 18-month window before the index date. There are in total 17,081 number of unique medical codes in this dataset.

The **MIMIC III** dataset (Johnson et al., 2016) is a publicly available dataset consisting of medical records of intensive care unit (ICU) patients over 11 years. We use a public query[1] to extract the binary mortality labels for the patients. Our goal is to use the codes in the patients' last visit to the ICU and predict their mortality outcome. Our dataset includes 46,520 patients out of whom 5810 have deceased (mortality=1). A totoal of 14,587 different medical codes are used in this dataset.

**Feature construction.** Given the sequence of visits $\mathbf{v}_1^{(i)}, \ldots, \mathbf{v}_T^{(i)}$ for patients $i = 1, \ldots, n$, we create a feature vector $\mathbf{x}_i \in \mathbb{N}_0^{|\mathcal{C}|}$ by counting the number of codes observed in the records of the $i$th patient. Given the large variations in the number of codes, we logarithmically bin the count data into 16 bins. The final data is in the form of $(\mathbf{x}_i, y_i)$ where $y_i$ is $i$th patient's label; heart failure and mortality outcome in the Sutter and MIMIC III datasets, respectively.

**Training details.** Given the fact that we generate synthetic datasets for training the causality detector neural networks, we can generate as many new batches of data for training and parameter tuning purposes as required. We report the test results on a dataset of size 10,000 data points. For training and parameter tuning of the neural network model in Section 2.2, we perform the common 75%/10%/15% training/validation/test splits. Details of training the latter neural network are given in Appendix B.2.

### 3.2 EVALUATING THE PREDICTIVE PREFORMANCE OF CAUSAL REGULARIZER

In order to characterize the performance of the proposed causal regularizer, we perform penalized logistic regression with the proposed regularizer and the commonly used $L_1$ regularizer. Table 1 shows the test accuracy of heart failure and mortality prediction in Sutter and MIMIC datasets, respectively. We have run each algorithm ten times and report the mean and standard deviation of the performance measures. As we can see, the proposed causal regularizer does not significantly hurt the predictive performance, whereas the two-step procedure significantly reduces the accuracy.

---

[1] https://github.com/MIT-LCP/mimic-code/blob/master/concepts/cookbook/mortality.sql

Table 1: Prediction accuracy results on two datasets. (mean±standard deviation)

| Algorithms | Sutter | | MIMIC III | |
|---|---|---|---|---|
| | AUC | $F_1$ | AUC | $F_1$ |
| Causal Logistic | $0.8289 \pm 0.0064$ | $0.4147 \pm 0.0192$ | $0.9772 \pm 0.0022$ | $0.7871 \pm 0.0097$ |
| $L_1$ Logistic | $0.8289 \pm 0.0054$ | $0.4109 \pm 0.0150$ | $0.9774 \pm 0.0022$ | $0.7869 \pm 0.0095$ |
| Two Step | $0.7276 \pm 0.0086$ | $0.2686 \pm 0.0134$ | $0.9515 \pm 0.0033$ | $0.6745 \pm 0.0106$ |

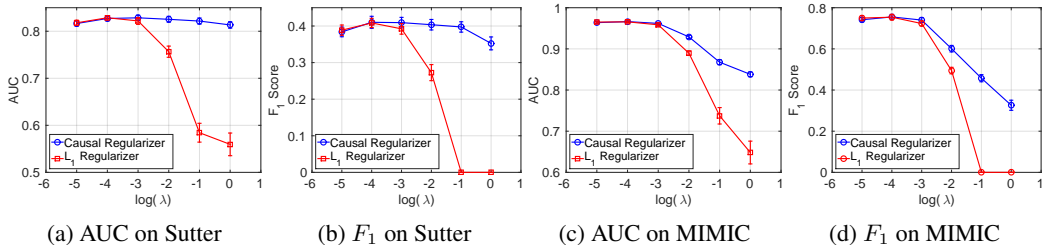

(a) AUC on Sutter (b) $F_1$ on Sutter (c) AUC on MIMIC (d) $F_1$ on MIMIC

Figure 4: Comparison of variable selection in logistic regression via the causal and $L_1$ regularizers on two datsets and two accuracy measures. Note the stability of variable selection by the causal regularizer as the penalization coefficient varies.

An interesting phenomenon, shown in Figure 4, is the relative robustness of the performance with respect to the value of the penalization parameter compared to the $L_1$ regularization case. This robustness comes at no surprise, because the causal regularizer assigns very small penalization coefficients to the causal variables and as we discussed in Section 2.2, only with very high values of penalization we can force all coefficients to become zero, see Figure 7 in Appendix A.1. Moreover, this robustness can be attributed to the fact that the causal regularizer might match the true generative process of the dataset better than the flat $L_1$ regularizer and puts the model under less pressure as we increase the penalization parameter. We demonstrate the predictive gain by the non-linear causal model in Figure 5a. Furthermore, the impact of changing the regularization parameter on the number of selected variables is visualized in Figure 8 in Appendix A.1.

## 3.3 EVALUATING THE CAUSAL PERFORMANCE OF CAUSAL REGULARIZER

In order to evaluate the performance of the algorithms in their ability to identify causal factors, we generate the top 100 influential factors by different methods. We ask a clinical expert to label each factor as "causal", "not-causal", and "potentially causal" and assign scores 1, 0, and 0.5 to them, respectively. Table 2 shows the average causality score by each algorithm based on the labels provided by the medical expert. As expected, $L_1$ regularized logistic regression performs poorly, as it is susceptible to the impact of confounded variables. Performance of the causally regularized logistic regression is superior to the two step procedure, which suggests that picking factors that are both causal and highly predictive leads to better causality score. This result together with the predictive results in Table 1 confirms that the causal regularizer can be efficiently used for finding few causal variables that are highly predictive of the target quantity.

The advantages of the regularized approach can also be seen by the results in Table 4. We have marked many disease codes that can potentially increase the risk of heart failure. However, the *predicted* causality score for them is lower than 0.5 and the two-step procedure would have eliminated from the predictors set (as shown in Table 10 in Appendix C). The causal regularizer approach is able to establish a balance between the prediction and causation and produce more plausible results.

## 3.4 EVALUATING THE MULTIVARIATE CAUSAL HYPOTHESES

We evaluate the performance of the proposed causal hypothesis generation against the case when we do not use any causal regularization. We generate two lists of top 50 hypotheses using two algorithms and ask our medical expert to label each hypothesis as causal, non-causal or possibly causal with corresponding scores of 1, 0, and 0.5. The results in Figure 5b shows that the causal regularizer can increase the causality score of the hypotheses by up to $20\%$. We also provide a qualitative analysis of

Table 2: Average causality score on the heart failure task computed using ground truth labels. For a higher resolution of number of top codes in the list see Figure 1a.

| # codes in the list | Causal Logistic | $L_1$ Logistic | Two Step |
|---|---|---|---|
| Top 20 | **0.725** | 0.400 | 0.425 |
| Top 50 | **0.520** | 0.330 | 0.450 |
| Top 100 | **0.485** | 0.315 | 0.375 |

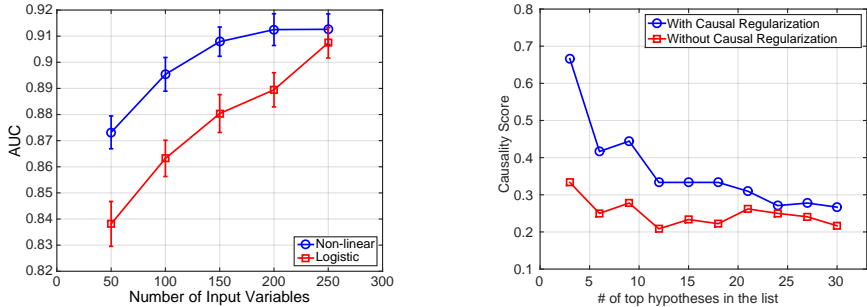

(a) Predictive gain by the non-linear model (b) Accuracy of causal hypothesis generation

Figure 5: (a) The predictive gain by the nonlinear causal model in Eq. (2) on the MIMIC III datset. The gain is more visible when fewer features are used in the analysis because the input become more expressive by themselves. We select the variables in the descending order of variance. (b) Average causality score computed using ground truth causality labels for generated hypotheses. We compute the score for top $k$ hypotheses reported by two algorithms.

Table 3: Examples of multivariate causal hypotheses generated via causal regularizer.

| Name | Conditions | Description |
|---|---|---|
| Aortic Dissection from Trauma | Dissection of aorta<br>Burn in multiple sites of trunk<br>Abdominal pain, lower left quadrant | This collection of diagnoses is is especially causal for heart failure, as heart failure can manifest as a complication of dissection of aorta. Dissection of aorta can present with abdominal pain, and may happen in traumatic injuries that involve burn of unspecified degree of other and multiple sites of trunk, occurring together. |
| Kidney Neoplasm and Severe Infections | Malignant neoplasm of kidney<br>History of infectious and parasitic diseases<br>Tuberculosis of lung | Neoplasms in the kidney may lead to paraneoplastic systemic effects that may lead to heart failure. Furthermore, having concurrent severe infections such as tuberculosis can also increase the risk of heart failure. |
| Metabolic Syndrome with Concurrent Infections and Pregnancy | Metabolic syndrome<br>Tuberculosis of lung<br>Obstetrical pulmonary embolism | Metabolic syndrome co-occurring with severe infections such as tuberculosis can lead to heart failure. Obstetrical pulmonary embolisms can lead to acute heart failure. |

the causal hypotheses generated by our algorithm. To this end, we pick several hypotheses and show that clinically they are meaningful. Three examples of multivariate causal hypotheses generated via causal regularizer are shown in Table 3.

## 4 CONCLUSION

We addressed the problem of exploring the high-dimensional causal hypothesis space in applications such as healthcare. We designed a causal regularizer that steers predictive algorithms towards explanations "as causal as possible". The proposed causal regularizer, based on our causality detector, does not increase the computational complexity of the $L_1$ regularizer and can be seamlessly integrated with a neural network to perform non-linear causality analysis. We also demonstrated the application of the proposed causal regularizer in generating multivariate causal hypotheses. Finally, we demonstrated the usefulness of the causal regularizer in detecting the causes of heart failure using an electronic health records dataset.

### ACKNOWLEDGMENT

The authors would like to thank Frederick Eberhardt for helpful discussions. Mohammad Taha Bahadori acknowledges the previous discussions with David C. Kale and Micheal E. Hankin on the concept of causal regularizer.

Table 4: Top 30 codes with causal regularization. The coefficient is $w_i$ from Eq. (1). The causality score in this table is the output of causality classifier.

| Code | Description | Coefficient | Causality |
|------|-------------|-------------|-----------|
| 794.31 | Nonspecific abnormal electrocardiogram [ECG] [EKG] | 0.3422 | 0.9351 |
| **425.8** | **Cardiomyopathy in other diseases classified elsewhere** | **0.3272** | **0.2322** |
| 786.05 | Shortness of breath | 0.3124 | 0.5536 |
| **424.90** | **Endocarditis, valve unspecified, unspecified cause** | **0.3086** | **0.3908** |
| **425.4** | **Other primary cardiomyopathies** | **0.2880** | **0.1351** |
| 427.9 | Cardiac dysrhythmia, unspecified | 0.2531 | 0.9864 |
| 785.9 | Other symptoms involving cardiovascular system | 0.2377 | 0.8024 |
| 585.6 | End stage renal disease | 0.2225 | 0.3948 |
| 511.9 | Unspecified pleural effusion | 0.2218 | 0.0839 |
| 425.9 | Secondary cardiomyopathy, unspecified | 0.2203 | 0.8024 |
| 782.3 | Edema | 0.2065 | 0.0027 |
| **278.01** | **Morbid obesity** | **0.1955** | **0.0345** |
| **424.0** | **Mitral valve disorders** | **0.1948** | **0.0003** |
| 427.31 | Atrial fibrillation | 0.1762 | 1.0000 |
| **410.90** | **Acute myocardial infarction of unspecified site, episode of care unspecified** | **0.1756** | **0.2510** |
| 426.3 | Other left bundle branch block | 0.1690 | 0.4890 |
| **424.1** | **Aortic valve disorders** | **0.1649** | **0.0012** |
| 879.8 | Open wound(s) (multiple) of unspecified site(s), without mention of complication | 0.1645 | 0.6399 |
| 429.3 | Cardiomegaly | 0.1619 | 0.5022 |
| 780.60 | Fever, unspecified | 0.1602 | 0.7747 |
| 482.9 | Bacterial pneumonia, unspecified | 0.1514 | 0.7482 |
| 786.09 | Other respiratory abnormalities | 0.1454 | 0.7305 |
| 496 | Chronic airway obstruction, not elsewhere classified | 0.1403 | 0.9990 |
| V42.0 | Kidney replaced by transplant | 0.1398 | 0.4351 |
| 250.03 | Diabetes mellitus without mention of complication, type I [juvenile type], uncontrolled | 0.1388 | 0.4727 |
| 276.51 | Dehydration | 0.1347 | 0.6738 |
| 403.10 | Hypertensive chronic kidney disease, benign, with chronic kidney disease stages I $\sim$ IV | 0.1316 | 0.7488 |
| **250.50** | **Diabetes with ophthalmic manifestations, type II or unspecified type, not uncontrolled** | **0.1283** | **0.2271** |
| 427.89 | Other specified cardiac dysrhythmias | 0.1282 | 0.9416 |
| 250.51 | Diabetes with ophthalmic manifestations, type I [juvenile type], not stated as uncontrolled | 0.1234 | 0.5473 |

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

## A    THE SAMPLING PROCEDURE FOR COUNT INDEPENDENT VARIABLES

As described in Section 3, our independent variables have count data type. Thus, we need to generate data from distributions for count data, such as Poisson or Zipf distributions with fixed support size of 16. Looking at the histogram of maximum number of code occurrences in Figure 6, we observe that many codes only occur at most once or twice. Thus, we also generate binary and trinary distributions from flat Dirichlet distributions. Finally, to make sure that the space is fully spanned, we also generate samples from Dirichlet distribution with 16 categories. In summary, the $\mathrm{dist}(s, K)$ is the mixture of these five distributions. The parameters of Poisson and Zipf are sampled from $\chi^2(1)$ distribution.

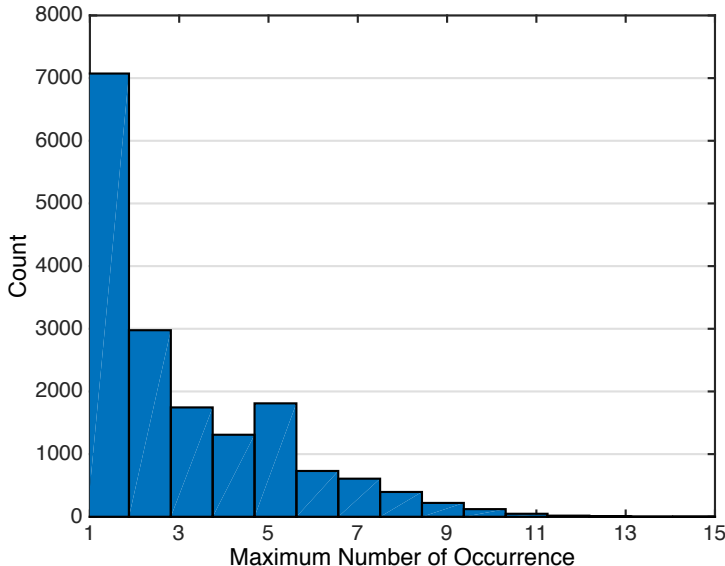

Figure 6: Histogram of maximum number of code occurrences.

Let $\mathrm{dist}(s, K)$ denotes a discrete distribution with parameter $s$ and given support size $K$.

**Direct** $X \to Y$:
1. Sample $s \sim \chi^2(2)$. Generate $P_X = \mathrm{dist}(s, K)$.
2. Sample $P_{Y|X} \sim \mathrm{Unif}(0, 1)$ for $K$ times.
3. Compute the $2K$-dimensional vector
$\mathbf{P}_{X,Y}(x, y) = [p(1, 0), \dots, p(K, 0), p(1, 1), \dots, p(K, 1)]$.

Algorithm 2: Another example of generating the synthetic dataset.

Sampling from the graphical models in Figure 2 is done writing the factorization and sampling from directed edges and finally marginalization with respect to hidden variables (Wainwright & Jordan, 2008). The hidden variables are selected to be categorical variables with cardinality selected uniformly from the integers in the interval $[2, 100]$. The conditional distribution of the hidden variables is selected to be Dirichlet distribution with all-ones parameter vector.

### A.1    EVALUATING THE CAUSALITY DETECTOR

Table 5 show two advantages of the proposed sampling procedure for count data in Appendix A in comparison to the binary case proposed by Chalupka et al. (2016). First, in the synthetic dataset, the test error is significantly lower. This is because the size of input to the neural causality detector is 32 compared to 4 for the binary case. Applying the causality detectors to our data, we observe that the causality scores generated by our sampling scheme has significantly higher correlation with the mutual information between independent variables and the target label. Figure 10 in Appendix C

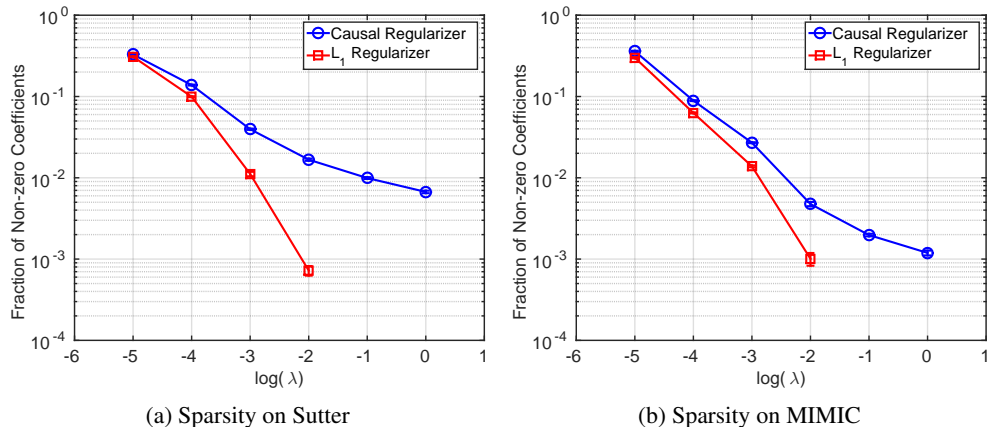

(a) Sparsity on Sutter (b) Sparsity on MIMIC

Figure 7: Comparison of variable selection in logistic regression via the causal and $L_1$ regularizers on two datsets and two accuracy measures. Note the stability of variable selection by the causal regularizer as the penalization coefficient varies.

highlights another advantage of the sampling procedure for count data as it is able to identify a larger portion of the variables as non-causal, which is more inline the expectations. Table 7 in Appendix C shows that the mutual information identifies V70.0 (Routine general medical examination at a health care facility) as highly correlated, but the causality detector correctly identifies it as non-causal with causality score 0.0000.

In particular, in Figure 9, the Spearman's rank correlation is $\rho = 0.6689$ which indicates a strong correlation. This is intuitive as we expect on average the causal connections to create stronger correlations. Another consequence of the large correlation makes regularization by the non-causality scores safer and guarantees that it will not significantly hurt the predictive performance. In Figure 9, we have marked four codes in the four corner of the figure. An example of highly correlated and causal code we can point out 250.00 (Diabetes mellitus without mention of complication) which is a known cause of heart failure. Code 362.01 (Background diabetic retinopathy) is an effect of diabetes —a common cause of heart failure. Code V06.5 (Need for TD vaccination) is an example of neither causal nor correlated code. Finally, code 365.00 (Preglaucoma) is known for increasing the risk of heart failure, despite the fact that it is not very correlated with heart failure.

## B DETAILS OF THE NEURAL NETWORKS

### B.1 NEURAL CAUSALITY DETECTOR ARCHITECTURE

We used a multilayer perceptron with seven layers of size 1024 with rectified linear units as activation functions. We use batch normalization for each layer. We used adamax for training and early stopping based on the validation accuracy. Implementation is done in Theano 0.8.

### B.2 THE NEURAL NETWORK IN SECTION 2.2

After tuning the parameters of the neural network, we ended up using a multilayer perceptron with three layers with rectified linear units as activation functions. The embedding dimension was 128 obtained by training GloVe on the entire dataset. We used dropout with rate $p = 0.8$ and adadelta for optimization and early stopping based on the validation accuracy. Implementation is done in Theano 0.8.

## C QUALITATIVE RESULTS

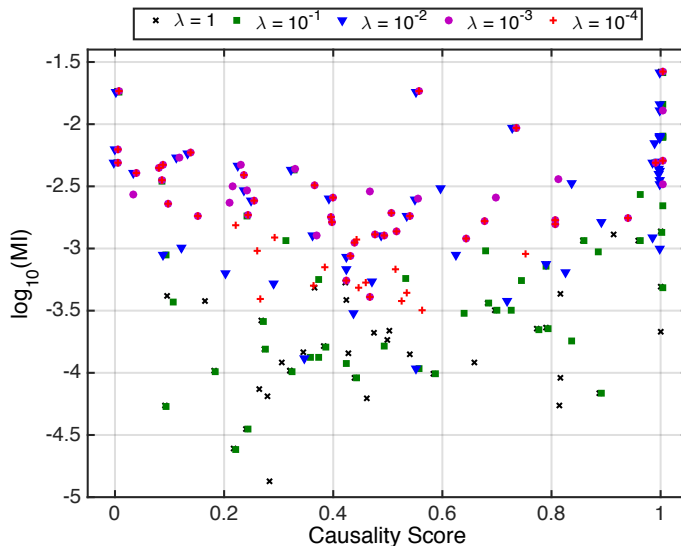

Figure 8: The impact of $\lambda$ on the top 50 selected variables, marked in the original plot in Figure 9 As $\lambda$ increases, there is a shift from left-up towards down-right corner and the trade-off shifts towards selecting more causal codes despite possibly lower mutual information. In this figure a small noise has been added to the points to visualize the overlapped points.

| Algorithm | Error Rate | Spearman Correlation w/ Mutual Information |
|---|---|---|
| Binary | 0.2165 | -0.0099 (0.4506) |
| Count | 0.0617 | 0.6689 (0.0000) |

Table 5: Summary of the results

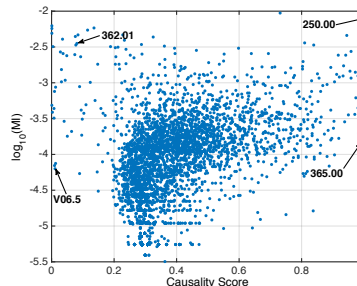

Figure 9: The scatter plot of causation score vs. mutual information.

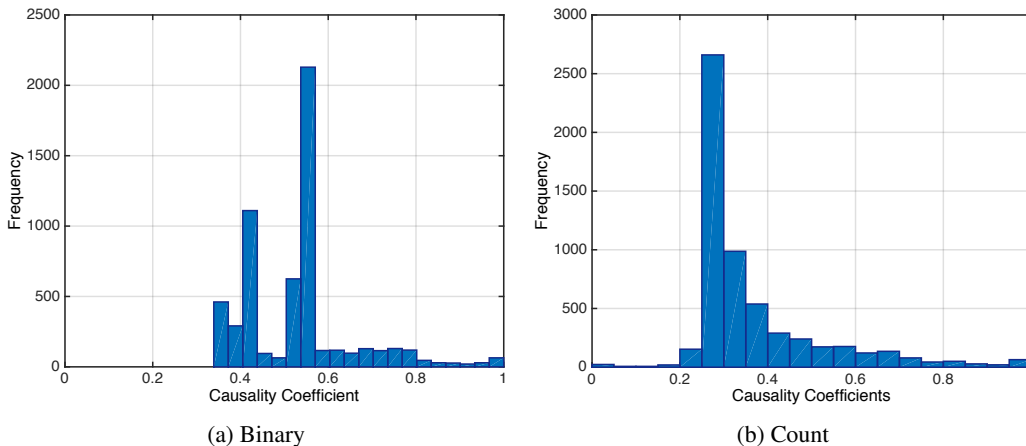

(a) Binary

(b) Count

Figure 10: Distribution of the coefficients generated by the causality detector.

Table 6: Performance of simple causality detector using count data. The causality score is the output of our neural network in Section 2.1 $\mathbb{P}[X_i \to Y]$. $MI$ denotes the mutual information between $X_i$ and $Y$.

| Code | Description | $\log(MI)$ | Causality Score |
|---|---|---|---|
| V58.83 | Encounter for therapeutic drug monitoring | -2.1050 | 1.0000 |
| 250.00 | Diabetes mellitus without mention of complication, type II or unspecified type, not stated as uncontrolled | -2.0962 | 1.0000 |
| 401.9 | Unspecified essential hypertension | -2.1023 | 1.0000 |
| V58.61 | Long-term (current) use of anticoagulants | -1.8371 | 1.0000 |
| 427.31 | Atrial fibrillation | -1.5795 | 1.0000 |
| 780.52 | Insomnia, unspecified | -3.3101 | 1.0000 |
| 272.4 | Other and unspecified hyperlipidemia | -2.6500 | 1.0000 |
| 401.1 | Benign essential hypertension | -2.4455 | 1.0000 |
| 702.0 | Actinic keratosis | -3.1493 | 1.0000 |
| 530.81 | Esophageal reflux | -3.3658 | 1.0000 |
| V04.81 | Need for prophylactic vaccination and inoculation against influenza | -3.6712 | 1.0000 |
| V10.83 | Personal history of other malignant neoplasm of skin | -3.4214 | 1.0000 |
| 250.60 | Diabetes with neurological manifestations, type II or unspecified type, not stated as uncontrolled | -2.3711 | 0.9999 |
| 414.00 | Coronary atherosclerosis of unspecified type of vessel, native or graft | -1.8896 | 0.9999 |
| 250.02 | Diabetes mellitus without mention of complication, type II or unspecified type, uncontrolled | -2.4819 | 0.9999 |
| 733.00 | Osteoporosis, unspecified | -3.4939 | 0.9999 |
| 692.74 | Other chronic dermatitis due to solar radiation | -2.9428 | 0.9999 |
| 244.9 | Unspecified acquired hypothyroidism | -3.1736 | 0.9999 |
| 311 | Depressive disorder, not elsewhere classified | -2.8482 | 0.9999 |
| 714.0 | Rheumatoid arthritis | -3.0010 | 0.9995 |
| 724.2 | Lumbago | -3.4554 | 0.9995 |
| 493.90 | Asthma, unspecified type, unspecified | -2.8824 | 0.9994 |
| V43.1 | Lens replaced by other means | -2.8667 | 0.9994 |
| 250.40 | Diabetes with renal manifestations, type II or unspecified type, not stated as uncontrolled | -2.3952 | 0.9990 |
| 496 | Chronic airway obstruction, not elsewhere classified | -2.2993 | 0.9990 |
| 786.2 | Cough | -2.3533 | 0.9987 |
| 173.3 | — | -4.1204 | 0.9964 |
| 733.90 | Disorder of bone and cartilage, unspecified | -3.4001 | 0.9919 |
| 786.50 | Chest pain, unspecified | -2.5651 | 0.9915 |
| 285.9 | Anemia, unspecified | -2.1526 | 0.9905 |

Table 7: Top 20 codes with highest mutual information with the heart failure outcome. The causality score is the output of our neural network in Section 2.1 $\mathbb{P}[X_i \rightarrow Y]$. $MI$ denotes the mutual information between $X_i$ and $Y$.

| Code | Description | $\log(MI)$ | Causality Score |
|---|---|---|---|
| 782.3 | Edema | -1.7355 | 0.0027 |
| 424.1 | Aortic valve disorders | -2.2021 | 0.0012 |
| 425.4 | Other primary cardiomyopathies | -2.2330 | 0.1351 |
| **V70.0** | **Routine general medical examination at a health care facility** | **-2.2420** | **0.0000** |
| 443.9 | Peripheral vascular disease, unspecified | -2.2668 | 0.1145 |
| 424.0 | Mitral valve disorders | -2.3088 | 0.0003 |
| 250.50 | Diabetes with ophthalmic manifestations, type II or unspecified type, not stated as uncontrolled | -2.3288 | 0.2271 |
| 511.9 | Unspecified pleural effusion | -2.3320 | 0.0839 |
| 427.32 | Atrial flutter | -2.3508 | 0.0767 |
| 278.01 | Morbid obesity | -2.3924 | 0.0345 |
| 425.8 | Cardiomyopathy in other diseases classified elsewhere | -2.4090 | 0.2322 |
| 362.01 | Background diabetic retinopathy | -2.4536 | 0.0815 |
| 584.9 | Acute kidney failure, unspecified | -2.4661 | 0.2882 |
| 412 | Old myocardial infarction | -2.4750 | 0.0780 |
| 428.0 | Congestive heart failure, unspecified | -2.4946 | 0.0039 |
| 791.0 | Proteinuria | -2.4984 | 0.1883 |
| 357.2 | Polyneuropathy in diabetes | -2.5040 | 0.2120 |
| 402.90 | Unspecified hypertensive heart disease without heart failure | -2.5103 | 0.2034 |
| 250.42 | Diabetes with renal manifestations, type II or unspecified type, uncontrolled | -2.5310 | 0.2378 |
| 280.9 | Iron deficiency anemia, unspecified | -2.5661 | 0.0283 |
| 427.1 | Paroxysmal ventricular tachycardia | -2.5884 | 0.0367 |
| **V53.31** | **Fitting and adjustment of cardiac pacemaker** | **-2.6014** | **0.2894** |
| 459.81 | Venous (peripheral) insufficiency, unspecified | -2.6093 | 0.0748 |
| 410.90 | Acute myocardial infarction of unspecified site, episode of care unspecified | -2.6154 | 0.2510 |
| 588.81 | Secondary hyperparathyroidism (of renal origin) | -2.6199 | 0.0478 |
| 250.62 | Diabetes with neurological manifestations, type II or unspecified type, uncontrolled | -2.6367 | 0.2051 |
| 414.8 | Other specified forms of chronic ischemic heart disease | -2.6403 | 0.0923 |
| 362.02 | Proliferative diabetic retinopathy | -2.6490 | 0.2629 |
| 586 | Renal failure, unspecified | -2.7324 | 0.2388 |
| 250.52 | Diabetes with ophthalmic manifestations, type II or unspecified type, uncontrolled | -2.7375 | 0.2154 |

Table 8: Top 20 causal codes ($c_i \geq 0.5$) with highest mutual information. The causality score is the output of our neural network in Section 2.1 $\mathbb{P}[X_i \to Y]$. $MI$ denotes the mutual information between $X_i$ and $Y$.

| Code | Description | log($MI$) | Causality Score |
|---|---|---|---|
| 427.31 | Atrial fibrillation | -1.5795 | 1.0000 |
| V58.61 | Long-term (current) use of anticoagulants | -1.8371 | 1.0000 |
| 414.00 | Coronary atherosclerosis of unspecified type of vessel, native or graft | -1.8896 | 0.9999 |
| 250.00 | Diabetes mellitus without mention of complication, type II or unspecified type, not stated as uncontrolled | -2.0962 | 1.0000 |
| 401.9 | Unspecified essential hypertension | -2.1023 | 1.0000 |
| V58.83 | Encounter for therapeutic drug monitoring | -2.1050 | 1.0000 |
| 285.9 | Anemia, unspecified | -2.1526 | 0.9905 |
| 496 | Chronic airway obstruction, not elsewhere classified | -2.2993 | 0.9990 |
| 427.9 | Cardiac dysrhythmia, unspecified | -2.3100 | 0.9864 |
| 585.3 | Chronic kidney disease, Stage III (moderate) | -2.3406 | 0.9447 |
| 786.2 | Cough | -2.3533 | 0.9987 |
| 250.60 | Diabetes with neurological manifestations, type II or unspecified type, not stated as uncontrolled | -2.3711 | 0.9999 |
| 250.40 | Diabetes with renal manifestations, type II or unspecified type, not stated as uncontrolled | -2.3952 | 0.9990 |
| 585.4 | Chronic kidney disease, Stage IV (severe) | -2.4419 | 0.8091 |
| 401.1 | Benign essential hypertension | -2.4455 | 1.0000 |
| 285.21 | Anemia in chronic kidney disease | -2.4734 | 0.8384 |
| 250.02 | Diabetes mellitus without mention of complication, type II or unspecified type, uncontrolled | -2.4819 | 0.9999 |
| 786.50 | Chest pain, unspecified | -2.5651 | 0.9915 |
| 703.8 | Other specified diseases of nail | -2.5659 | 0.9583 |
| 427.89 | Other specified cardiac dysrhythmias | -2.6153 | 0.9416 |
| 780.79 | Other malaise and fatigue | -2.6174 | 0.9169 |
| 272.4 | Other and unspecified hyperlipidemia | -2.6500 | 1.0000 |
| 250.01 | Diabetes mellitus without mention of complication, type I [juvenile type], not stated as uncontrolled | -2.7045 | 0.8049 |
| 794.31 | Nonspecific abnormal electrocardiogram [ECG] [EKG] | -2.7602 | 0.9351 |
| 785.9 | Other symptoms involving cardiovascular system | -2.7719 | 0.8024 |
| 724.02 | Spinal stenosis, lumbar region, without neurogenic claudication | -2.7874 | 0.8932 |
| 425.9 | Secondary cardiomyopathy, unspecified | -2.8071 | 0.8024 |
| 274.9 | Gout, unspecified | -2.8380 | 0.9539 |
| 311 | Depressive disorder, not elsewhere classified | -2.8482 | 0.9999 |
| 10000 | — | -2.8504 | 0.9721 |

Table 9: Top 30 codes with causal regularization. The coefficient is $w_i$ from Eq. (1).

| Code | Description | Coefficient | Causality Score |
|---|---|---|---|
| 794.31 | Nonspecific abnormal electrocardiogram [ECG] [EKG] | 0.3422 | 0.9351 |
| 425.8 | Cardiomyopathy in other diseases classified elsewhere | 0.3272 | 0.2322 |
| 786.05 | Shortness of breath | 0.3124 | 0.5536 |
| 424.90 | Endocarditis, valve unspecified, unspecified cause | 0.3086 | 0.3908 |
| 425.4 | Other primary cardiomyopathies | 0.2880 | 0.1351 |
| 427.9 | Cardiac dysrhythmia, unspecified | 0.2531 | 0.9864 |
| 785.9 | Other symptoms involving cardiovascular system | 0.2377 | 0.8024 |
| 585.6 | End stage renal disease | 0.2225 | 0.3948 |
| 511.9 | Unspecified pleural effusion | 0.2218 | 0.0839 |
| 425.9 | Secondary cardiomyopathy, unspecified | 0.2203 | 0.8024 |
| 782.3 | Edema | 0.2065 | 0.0027 |
| 278.01 | Morbid obesity | 0.1955 | 0.0345 |
| 424.0 | Mitral valve disorders | 0.1948 | 0.0003 |
| 427.31 | Atrial fibrillation | 0.1762 | 1.0000 |
| 410.90 | Acute myocardial infarction of unspecified site, episode of care unspecified | 0.1756 | 0.2510 |
| 426.3 | Other left bundle branch block | 0.1690 | 0.4890 |
| 424.1 | Aortic valve disorders | 0.1649 | 0.0012 |
| 879.8 | Open wound(s) (multiple) of unspecified site(s), without mention of complication | 0.1645 | 0.6399 |
| 429.3 | Cardiomegaly | 0.1619 | 0.5022 |
| 780.60 | Fever, unspecified | 0.1602 | 0.7747 |
| 482.9 | Bacterial pneumonia, unspecified | 0.1514 | 0.7482 |
| 786.09 | Other respiratory abnormalities | 0.1454 | 0.7305 |
| 496 | Chronic airway obstruction, not elsewhere classified | 0.1403 | 0.9990 |
| V42.0 | Kidney replaced by transplant | 0.1398 | 0.4351 |
| 250.03 | Diabetes mellitus without mention of complication, type I [juvenile type], uncontrolled | 0.1388 | 0.4727 |
| 276.51 | Dehydration | 0.1347 | 0.6738 |
| 403.10 | Hypertensive chronic kidney disease, benign, with chronic kidney disease stage I through stage IV, or unspecified | 0.1316 | 0.7488 |
| 250.50 | Diabetes with ophthalmic manifestations, type II or unspecified type, not stated as uncontrolled | 0.1283 | 0.2271 |
| 427.89 | Other specified cardiac dysrhythmias | 0.1282 | 0.9416 |
| 250.51 | Diabetes with ophthalmic manifestations, type I [juvenile type], not stated as uncontrolled | 0.1234 | 0.5473 |

Table 10: Top 30 codes with the two-step procedure. The coefficient is the logistic regression coefficient with $L_1$ regularization after causal variable selection.

| Code | Description | Coefficient | Causality Score |
|---|---|---|---|
| 786.05 | Shortness of breath | 0.3620 | 0.5536 |
| 427.9 | Cardiac dysrhythmia, unspecified | 0.2501 | 0.9864 |
| 794.31 | Nonspecific abnormal electrocardiogram [ECG] [EKG] | 0.2100 | 0.9351 |
| 427.31 | Atrial fibrillation | 0.2017 | 1.0000 |
| 785.9 | Other symptoms involving cardiovascular system | 0.1781 | 0.8024 |
| 786.09 | Other respiratory abnormalities | 0.1772 | 0.7305 |
| 414.00 | Coronary atherosclerosis of unspecified type of vessel, native or graft | 0.1478 | 0.9999 |
| 496 | Chronic airway obstruction, not elsewhere classified | 0.1465 | 0.9990 |
| 707.15 | Ulcer of other part of foot | 0.1407 | 0.5519 |
| 707.14 | Ulcer of heel and midfoot | 0.1383 | 0.5113 |
| 879.8 | Open wound(s) (multiple) of unspecified site(s), without mention of complication | 0.1334 | 0.6399 |
| 682.9 | Cellulitis and abscess of unspecified sites | 0.1328 | 0.5989 |
| 585.4 | Chronic kidney disease, Stage IV (severe) | 0.1322 | 0.8091 |
| 414.01 | Coronary atherosclerosis of native coronary artery | 0.1253 | 0.6930 |
| 250.51 | Diabetes with ophthalmic manifestations, type I [juvenile type], not stated as uncontrolled | 0.1180 | 0.5473 |
| 285.9 | Anemia, unspecified | 0.1172 | 0.9905 |
| 250.02 | Diabetes mellitus without mention of complication, type II or unspecified type, uncontrolled | 0.1172 | 0.9999 |
| 425.9 | Secondary cardiomyopathy, unspecified | 0.1132 | 0.8024 |
| 250.40 | Diabetes with renal manifestations, type II or unspecified type, not stated as uncontrolled | 0.1040 | 0.9990 |
| 427.89 | Other specified cardiac dysrhythmias | 0.1024 | 0.9416 |
| 429.3 | Cardiomegaly | 0.0968 | 0.5022 |
| 785.0 | Tachycardia, unspecified | 0.0828 | 0.5367 |
| 724.02 | Spinal stenosis, lumbar region, without neurogenic claudication | 0.0814 | 0.8932 |
| 482.9 | Bacterial pneumonia, unspecified | 0.0796 | 0.7482 |
| 786.50 | Chest pain, unspecified | 0.0789 | 0.9915 |
| 250.00 | Diabetes mellitus without mention of complication, type II or unspecified type, not stated as uncontrolled | 0.0760 | 1.0000 |
| V43.3 | Heart valve replaced by other means | 0.0749 | 0.8558 |
| 434.91 | Cerebral artery occlusion, unspecified with cerebral infarction | 0.0688 | 0.6249 |
| 285.21 | Anemia in chronic kidney disease | 0.0684 | 0.8384 |
| V72.84 | Pre-operative examination, unspecified | 0.0629 | 0.9866 |

