# Peer review of "Neural Causal Regularization under the Independence of Mechanisms Assumption"

_ICLR 2017 — rejected_

[Public Comment · David Lopez-Paz · 07 Nov 2016]
**Novelty and prior work**

Learning classifiers to distinguish between causal, anticausal, and confounded dependencies is not a novel idea. It was introduced in the competitions of Guyon [1,2], and later developed in the context of kernel machines [3,4], learning theory [4], and neural networks [5]. These classifiers were applied to discrete data [1,2,3,4,5,6], extend to multiple random variables [4 Appendix C, 6], generalize the additive noise model assumption [5,6], consider hidden confounding [1,2,4 Appendix C, 6], and make use of classifier soft-scores to measure predictive uncertainty [1,2,3,4,5,6].

The idea of causal regularization is not novel, either. It was proposed at least in [1 Page 181] and [2 Section 3]. Causal regularization is also at the heart of invariant prediction [7], and causal transfer [8,9].

None of these works are cited by this manuscript or by the companion manuscript (Chalupka et al., 2016). This omission undermines the novelty claims of the manuscript, and prevents the readers from easily seeing what the contribution is. The authors should therefore add a comprehensive discussion of the related works and a realistic discussion of the claimed novelty in this work.

[1]

[Reviewer Comment · AnonReviewer1 · 02 Dec 2016]
**Reproducibility**

While the paper provides enough detail to re-implement the proposed methods, there is no discussion as to whether the heart failure dataset will be released. This would be key for reproducibility.

[Official Review · AnonReviewer1 · rating 6 · confidence 5 · 14 Dec 2016]
**No Title**

The present submission discusses a "causal regularizer", which promotes the use of causal dependencies (X -> Y, where X is a feature of the learning problem, and Y is the target variable) in predictive models. Similarly, such causal regularizer penalizes the use of non-causal dependencies, which can arise due to reverse causation (Y -> X) or confounding (X <- Z -> Y, where Z is a hidden confounder).

+ Overall, this submission tackles one of the most important problems in machine learning, which is to build causal models. The paper discusses and addresses this issue effectively when applied to a dataset in heart disease. In their experiments, the authors correctly identify some of the common causes of heart disease by virtue of their causal regularizer.

- The authors do not discuss the robustness of their approach with respect to choice of hyper-parameters (both describing the neural network architecture and the generative model that synthesizes artificial causal data). This seems like a crucial issue, in particular when dealing with medical data.

- The conclusions of the experimental evaluation should be discussed in greater length. On the one hand, Figure 4.a shows that there are no differences between L1 and causal regularization in terms of predictive performance, but it is difficult to conclude if this result is statistically significant without access to error-bars. On the other hand, Table 3 describes the qualitative differences between L1 and causal regularization. However, this table is hard to read: How were the 30 rows selected? What does the red highlighting mean? Are these red rows some true causal features that were missed? If so, this is related to precision. What about recall? Did the causal regularization pick up many non-causal features as causal?

- Regarding causal classifiers, this paper should do a much better job at reviewing previous work. For instance, the paper "Towards a Learning Theory of Cause-Effect Inference" from Lopez-Paz et al. is missing from the references. However, this prior work studies many of the aspects that are hinted as novel in this submission. In particular, the prior work of Lopez-Paz 1) introduces the concept of Mother distribution (referred as Nature hyper-prior in this submission) which explicitly factorizes the distribution over causes and mechanisms, 2) circumvented intractable likelihoods by synthesizing and training on causal data, 3) tackled the confounding case (compare Figure 1 of this submission and Appendix C of Lopez-Paz), and 4) dealt with discrete data seamlessly (such as the ChaLearn data from Section 5.3 in Lopez-Paz).

On a positive note, this is a well-written paper that addresses the important, under-appreciated problem of incorporating causal reasoning into machine learning. On a negative note, the novelty of the technical contributions is modest and the qualitative evaluation of the results could be greatly extended. In short, I am leaning slightly towards acceptance.

[Official Review · AnonReviewer2 · rating 4 · confidence 4 · 17 Dec 2016]
**a good paper but a bit short of the mark in terms of presentation, methodology, and results**

The authors extend their method of causal discovery (Chalupka et al 2016) to include assumptions about sparsity via regularization.  They apply this extension to an interesting private dataset from Sutter Health.  While an interesting direction, I found the presentation somewhat confused, the methodological novelty smaller than the bulk of ICLR works, and the central results (or perhaps data; see below) inadequate to address questions of causality.

First, I found the presentation somewhat unclear.  The paper at some points seems to be entirely focused on healthcare data, at other points it uses it as a motivating example, and at other points it is neglected.  Also, algorithm 1 seems unreferenced, and I'm not entirely sure why it is needed.  Figure 2 is not needed for this community.  The key methodological advance in this work appears in section 2.1 (Causal regularizer), but it is introduced amidst toy examples and without clear terminology or standard methodological assumptions/build-up.  In Section 3.1 (bottom of first paragraph), key data and results seem to be relegated to the appendices.  Thus overall the paper read rather haphazardly.  Finally, there seems to be an assumption throughout of fairly intimate familiarity with the Cholupka preprint, which i think should be avoided.  This paper should stand alone.

Second, while the technical contributions/novelty are not a focus of the paper's presentation, I am concerned by the lack of methodological advance.  Essentially a regularization objective is added to the previous method, which of itself is not a bad idea, but I can't point to a technical novelty in the paper that the community can not do without.

Third, fundamentally i don't see how the experiments address the central question of causality; they show regularization behaving as expected (or rather, influencing weights as expected), but I don't think we really have any meaningful quantitative evidence that causality has been learned.  This was briefly discussed (see "ground truth causality?" and the response below).  I appreciate the technical challenges/impossibility of having such a dataset, but if that's the case, then I think this work is premature, since there is no way to really validate.

Overall it's clearly a sincere effort, but I found it wanting in terms of a few critical areas.

[Official Review · AnonReviewer3 · rating 5 · confidence 4 · 19 Dec 2016]
**Possibly practical method, but methodology and analysis fall short**

This paper proposes to use a causality score to weight a sparsity regularizer.  In that way, selected variables trade off between being causal and discriminative.  The framework is primarily evaluated on a proprietary health dataset.  While the dataset does give a good motivation to the problem setting, the paper falls a bit short for ICLR due to the lack of additional controlled experiments, relatively straightforward methodology (given the approach of Chalupka et al., arXiv Preprint, 2016, which is a more interesting paper from a technical perspective), and paucity of theoretical motivation.

At the core of this paper, the approach is effectively to weight a sparsity regularizer so that "causal" variables (as determined by a separate objective) are more likely to be selected.  This is generally a good idea, but we do not get a proper validation of this from the experiments as ground truth is absent.  A theorem on identifiability of causal+discriminative variables from a data sample combined with adequate synthetic experiments would have probably been sufficient, for example, to push the paper towards accept from a technical perspective, but as it is, it is lacking in insight and reproducibility.

[Author Response · Mohammad Taha Bahadori · 13 Jan 2017 (modified: 02 Feb 2017)]
**Major Revision**

We made a major revision to the paper.  Here is the summary of the main changes:

Experiments:
-- Providing **ground truth causality** evaluation (by hiring Robert Chen, a clinical expert). (Figure 1, Table 2) The results show a significant gain in causality discovery using the proposed method.
-- Including a new **publicly available dataset** (MIMIC III) to ensure reproducibility of the results. (Table 1 and Figure 4)
-- Adding standard deviation to the results.

Novel methodology:
-- Providing a neural network architecture that allows multivariate causal hypothesis generation, see Section 2.3 and Figure 2.b.
-- Providing the ground truth causality evaluation for the causal hypotheses generated by our algorithm. (Figure 5b)

Writeup:
-- Adding more discussion on the related works (Sections 2.1 and 2.2). 
---- In the introduction and Section 2.2 we clearly describe the novelty of this paper with respect to the papers listed by David Lopez-Paz and the reviewers.
-- Revising the background section 2.1 and making it both more accessible and more mathematically precise.
-- Adding new visualizations in Figure 2 and removing diagrams that didn’t help delivering the concepts.
-- Restructuring the methodology section. Now Section 2.2 only discusses the proposed causal regularizer and Section 2.3 describes its combination with neural networks.
-- Moving the experimental results on the causality detection (and other extra results) to Appendix A.1, to allow space for evaluation of the main results of the paper.
-- Finally, we revised the introduction accordingly.

[Final Decision · Program Chairs · 06 Feb 2017]
**ICLR committee final decision**

The reviewers pointed out several issues with the paper, and all recommended rejection. The revision seems to not have been enough to change their minds.